# Harmonic to anharmonic tuning of moiré potential leading to unconventional Stark effect and giant dipolar repulsion in $WS_2$/ $WSe_2$ heterobilayer

Suman Chatterjee [1,6], Medha Dandu[1,4,6], Pushkar Dasika[1,6], Rabindra Biswas[1], Sarthak Das[1,5], Kenji Watanabe [2], Takashi Taniguchi [3], Varun Raghunathan[1] & Kausik Majumdar [1] ✉

Excitonic states trapped in harmonic moiré wells of twisted heterobilayers is an intriguing testbed for exploring many-body physics. However, the moiré potential is primarily governed by the twist angle, and its dynamic tuning remains a challenge. Here we demonstrate anharmonic tuning of moiré potential in a $WS_2$/$WSe_2$ heterobilayer through gate voltage and optical power. A gate voltage can result in a local in-plane perturbing field with odd parity around the high-symmetry points. This allows us to simultaneously observe the first (linear) and second (parabolic) order Stark shift for the ground state and first excited state, respectively, of the moiré trapped exciton - an effect opposite to conventional quantum-confined Stark shift. Depending on the degree of confinement, these excitons exhibit up to twenty-fold gate-tunability in the lifetime (100 to 5 ns). Also, exciton localization dependent dipolar repulsion leads to an optical power-induced blueshift of ~1 meV/μW - a five-fold enhancement over previous reports.

Interlayer van der Waals interaction allows us to stack layers of transition metal dichalcogenides (TMDCs) onto each other with an arbitrary lattice mismatch[1–3]. This leads to an additional degree of freedom, the twist angle ($\theta$) between two successive layers, that governs the moiré pattern arising in the corresponding superlattice[4–7]. The lattice constant of the moiré superlattice is given by $a_M \approx \frac{a}{\sqrt{\theta^2 + \delta^2}}$ where $\delta$ is the lattice constant difference between the constituent monolayers and $a$ being the average lattice constant[6,8,9]. Different atomic registries present in this moiré superlattice (Fig. 1a) form a periodic potential fluctuation [$V_M(\mathbf{r})$] resulting from local strain and interlayer coupling[10,11]. Varying twist angle can dramatically change the material properties,

drawing attention from the researchers in the recent past[8,12–14]. Moiré superlattice in TMDC heterobilayer has been widely explored including observation of neutral moiré exciton[4,15,16], moiré trion[17–19], single photon emission[20,21], and correlated states[5,22,23].

Due to type-II band alignment, $WS_2$/$WSe_2$ heterobilayer supports an ultrafast charge transfer[24,25] with electrons staying in the $WS_2$ conduction band, and holes in the $WSe_2$ valance band, forming interlayer exciton (ILE)[8,9] under optical excitation (Fig. 1b). The moiré wells behave as two-dimensional harmonic traps for the ILEs[4,26,27].

The depth of the exciton moiré potential is determined by the twist angle and the degree of lattice mismatch between the two heterobilayers. Hence, dynamic tuning of moiré potential remains a

[1]Department of Electrical Communication Engineering, Indian Institute of Science, Bangalore 560012, India. [2]Research Center for Functional Materials, National Institute for Materials Science, 1-1 Namiki, Tsukuba 305-044, Japan. [3]International Center for Materials Nanoarchitectonics, National Institute for Materials Science, 1-1 Namiki, Tsukuba 305-044, Japan. [4]Present address: Molecular Foundry, Lawrence Berkeley National Laboratory, Berkeley, CA 94720, USA. [5]Present address: Institute of Materials Research and Engineering (IMRE), Agency for Science, Technology and Research (A*STAR), Singapore 138634, Republic of Singapore. [6]These authors contributed equally: Suman Chatterjee, Medha Dandu, Pushkar Dasika. ✉e-mail: kausikm@iisc.ac.in

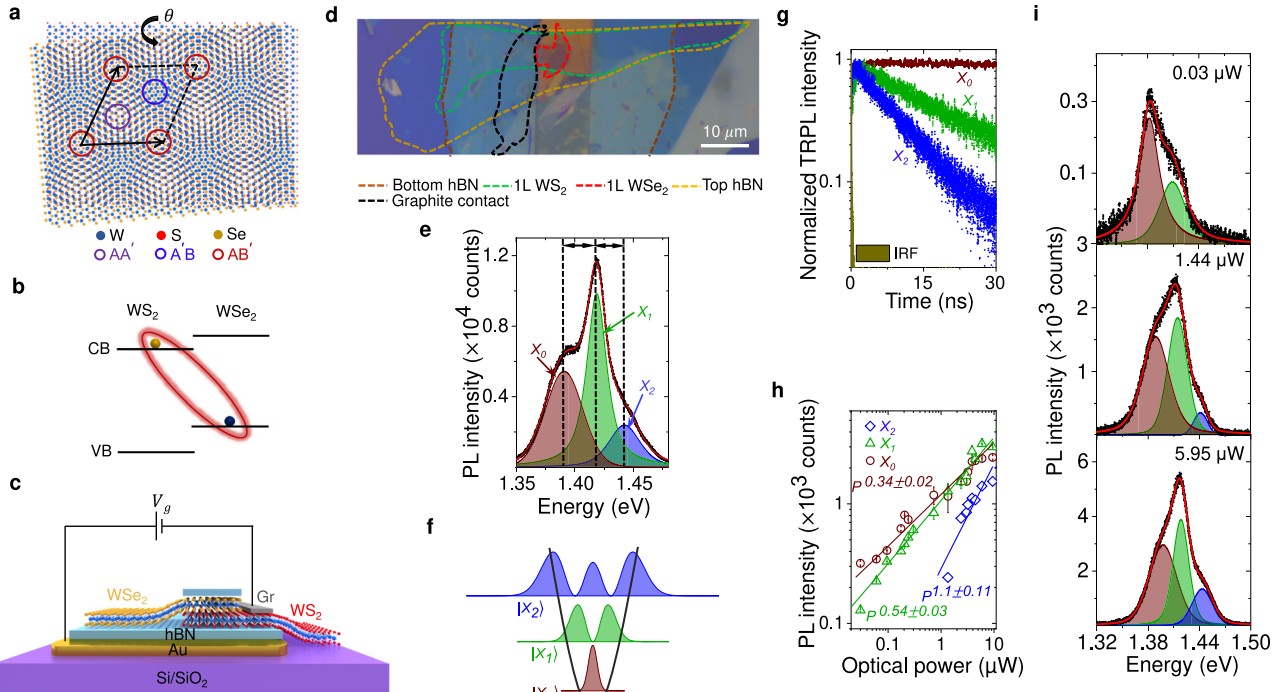

**Fig. 1 | Moiré trapped interlayer exciton. a** Different atomic registries in a twisted WS$_2$/WSe$_2$ bilayer with high symmetry points marked by colored circles. **b** Type-II heterojunction of WS$_2$/WSe$_2$ bilayer resulting in interlayer exciton. **c** Schematic of the heterobilayer with back gate connection. **d** Optical image of a fabricated device. The dotted colored lines indicate different flake boundaries. Scale bar is 10 μm. **e** Representative PL spectrum (using 532 nm CW laser) in the ILE regime (black symbols) and fitting (red trace) showing three clear ILE resonances denoted by $X_0$ (brown), $X_1$ (green), and $X_2$ (blue) at $V_g = 0$ V and $P = 0.675$ μW. Black arrows indicate near equal spacing. **f** Schematic representation of three ILE states in a harmonic moiré potential well with varying degree of localization. **g** Raw TRPL spectra along with IRF for the three ILE resonances showing varying decay time scales at $V_g = 0$ V ($P = 13.45$ μW), namely 100, 15, and 9.3 ns for $X_0$, $X_1$, and $X_2$, respectively. **h** Optical power dependent intensity plot (symbols) of the three ILEs in log-log scale following different power-laws (fitted by solid lines). **i** Evolution of power-dependent PL spectra (black symbols) at three different optical powers, along with fitting (red solid trace).

challenge, which, if realised, will be of great importance for both scientific exploration and applications. One could perturb the moiré potential by external stimulus, however, the perturbing potential may not necessarily be harmonic, breaking down the usual harmonic potential approximation for moiré well. In this work, we explore two such anharmonic perturbations to the WS$_2$/WSe$_2$ moiré potential well: the first one through a gate voltage which introduces anharmonic perturbation through screening at high doping regime; and the second one is through optical excitation which introduces the perturbing potential through ILE dipolar repulsion. In both cases, the harmonic to anharmonic switching of the moiré potential manifests through a corresponding change from an equal to unequal inter-excitonic spectral separation. In such a scenario, we explore several intriguing features of the moiré excitons, including giant lifetime tunability, anomalous Stark shift, and dipolar repulsion induced large spectral blueshift.

## Results and discussion

We prepare hBN-capped ~59° twisted (confirmed by second harmonic generation (SHG) spectroscopy in Supplementary Note 1 and Fig. 1) WS$_2$/WSe$_2$ heterobilayer (sample D1) with a back gate (see "Methods" section for sample preparation). The schematic and the optical image of sample D1 are illustrated in Fig. 1c and d. This twist angle creates a moiré superlattice with a lattice constant ~7.3 nm. Figure 1e shows a representative photoluminescence (PL) spectrum from the sample with 532 nm excitation at 4 K. The emission spectrum exhibits three separate, strong interlayer moiré excitonic resonances[28] $X_0$, $X_1$, and $X_2$ at ≈ 1.392, 1.418, and 1.442 eV, respectively (marked by black dashed line). The peaks exhibit alternating sign of the degree of circular

polarization (DOCP) (Supplementary Fig. 2), indicating the existence of moiré superlattice[4,6,29].

The near-equal inter-excitonic separation suggests that the three exciton resonances appear from excitonic states in the harmonic moiré potential well (Fig. 1f)[4,6,26,27]. This inter-excitonic separation can be tuned by varying the twist angle, which regulates the depth of the moiré potential well[4,30]. We verified this by measuring twist angle dependent PL spectra from three samples [D1 (-59°), D2 (-54°), and D3 (large angle misalignment)] in Supplementary Fig. 3. The time-resolved PL (TRPL) measurement (see "Methods" section) from sample D1 in Fig. 1g shows that the lifetime of the three species ($\tau_0 = 100$ ns, $\tau_1 = 15.3$ ns, and $\tau_2 = 9$ ns) increases significantly with stronger confinement. Accordingly, their PL intensity also exhibits significantly different power law with varying optical power (P): $I \propto P^{\alpha_i}$ with $\alpha_0 = 0.34 \pm 0.02$, $\alpha_1 = 0.59 \pm 0.03$, and $\alpha_2 = 1.1 \pm 0.11$ (Fig. 1h). The corresponding spectra at three different P values are shown in Fig. 1i. At low power (30 nW), $X_0$ emission is the dominant one, with negligible emission from $X_2$. However, at higher power (5.95 μW), three peaks are clearly discernable, and the fractional contribution of $X_0$ reduces, while $X_2$ emission becomes appreciable. All these observations indicate that the three different excitonic species correspond to moiré trapped excitonic states with varying degrees of localization (Fig. 1f). From the spectral separation between the quantized states, we calculate peak-to-peak moiré potential fluctuation of ≈150 meV (see Supplementary Note 2), as shown in Fig. 2c. Possible alternative explanations, such as phonon-sidebands and defect-bound excitons, are unlikely in our samples based on the observations including alternating signs of the DOCP and systematic tuning of the ILE peak separation with twist angle, doping, and optical power (discussed later).

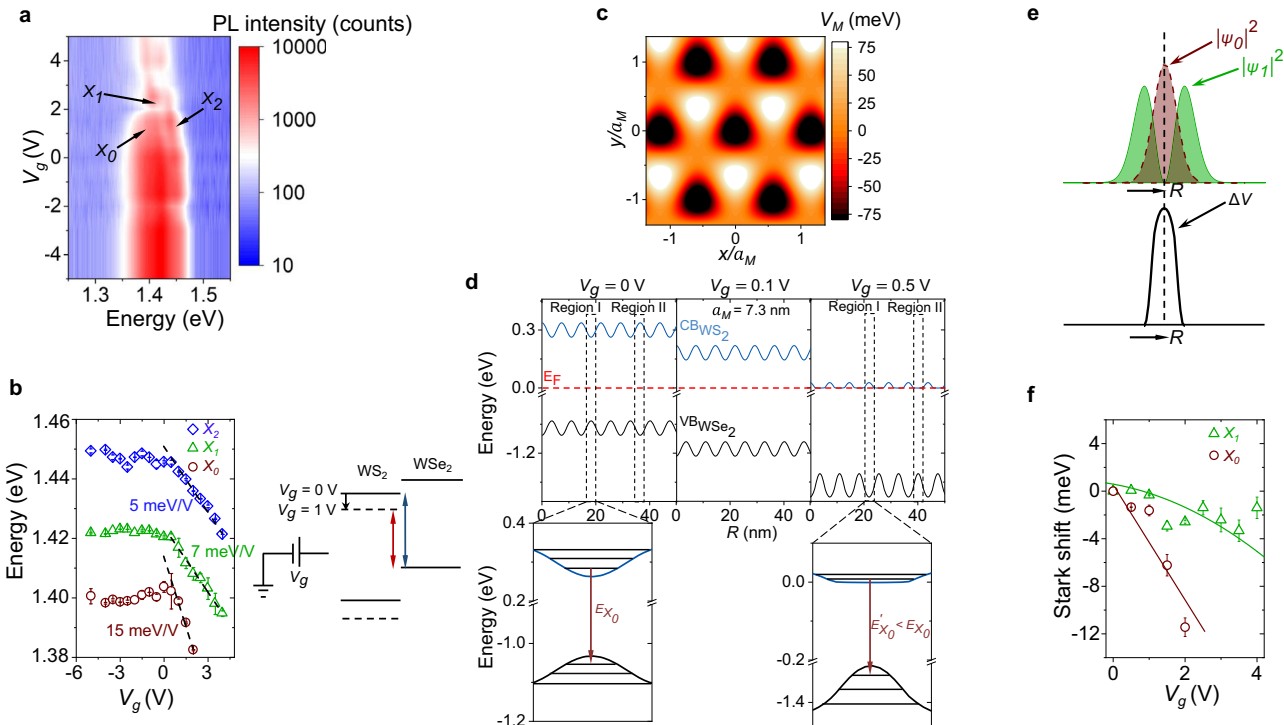

**Fig. 2 | Gate-tunable moiré potential and unconventional Stark effect. a** Color plot of $V_g$ dependent PL spectra showing $X_0$, $X_1$, and $X_2$ resonances. **b** Left panel: Fitted peak positions showing the gradual redshift of the three ILE peaks with $V_g$. The black dashed lines indicate guide-to-eye in the $V_g > 0$ regime. Right panel: interlayer bandgap reduction is shown schematically with increasing $V_g$. **c** 2D projection of the variation of the calculated moiré potential. **d** Top panel: Simulated conduction and valance band profile at three different $V_g$ (0, 0.1, and 0.5 V) values obtained by solving the Poisson equation with the moiré potential fluctuation (see Supplementary Note 3 for details). For simulation, the thickness of the gate dielectric (hBN) is assumed to be 20 nm. Region I (II) in the top left panel denotes the minimum (maximum) energy of the WS$_2$ conduction band due to moiré

potential induced spatial energy fluctuation. At lower $V_g$ (top middle panel), the conduction band gradually comes down in energy towards the Fermi level (red dashed line) maintaining the same degree of fluctuation. At higher $V_g$ (top right panel), when the conduction band is close to the Fermi level, it starts flattening due to screening. This also results in an enhancement in the valence band fluctuation. Bottom panel: Zoomed-in Region I at $V_g = 0$ V (in left) and $V_g = 0.5$ V (in right). The transition energy for $X_0$ ($E_{X_0}$, shown by arrow) decreases at higher $V_g$. (**e**) $|\psi_i|^2$ ($i = 0, 1$) plotted along with the in-plane perturbing potential $\Delta V$ indicating strong overlap (non-overlap) between $\Delta V$ and $|\psi_0|^2$ ($|\psi_1|^2$) due to different parity of the wave functions. **f** Stark shift of $X_0$ ($\delta_{X_0}$) and $X_1$ ($\delta_{X_1}$) plotted with $V_g$. $\delta_{X_0}$ ($\delta_{X_1}$) shows a linear(parabolic) Stark shift fitting (solid traces).

## Gate tunability

Figure 2a shows a color plot of the interlayer exciton emission spectra as a function of gate voltage ($V_g$). The estimated n-doping density at the highest applied $V_g$ (=5 V) is <$1.5 \times 10^{12}$ cm$^{-2}$ (see Supplementary Fig. 4). This is well below the moiré trap density ($N_M$) $\approx 2 \times 10^{12}$ cm$^{-2}$ for $a_M \sim 7.3$ nm. The fitted peak positions are shown in the left panel of Fig. 2b (see individual spectra in Supplementary Fig. 5). While the $V_g < 0$ V region is nearly featureless, $V_g > 0$ V (n-doping) region has three conspicuous features: (a) there is a reduction in emission intensity for all the three ILE peaks, with $X_0$ disappearing at high $V_g$; (b) there is a large and unequal redshift for the peaks for $V_g > 0$; and (c) the inter-excitonic separation changes at higher $V_g$, indicating induced anharmonicity. The reduction in emission intensity with an increase in $V_g$ rules out the charged excitonic (trion) nature of any of the three peaks. Figure 2b (right panel) schematically explains the origin of the strong redshift with $V_g$. At positive $V_g$, the WS$_2$ layer becomes n-doped. Due to small thermal energy at 4 K, the wave function of the induced electrons remains primarily in the WS$_2$ layer, with a fraction of it extends into the WSe$_2$ bandgap as an evanescent state with imaginary wave vector. Such a wave function distribution creates a screening of the gate field, and in turn a relative potential difference between WS$_2$ and WSe$_2$ layers, reducing the interlayer bandgap. Note that, the presence of the charge density from the evanescent state in WSe$_2$ is essential to create such relative potential difference between the two layers, else dictated by the self-consistent electrostatics, a zero induced charge density in WSe$_2$ layer would result in pinning of the

WSe$_2$ potential with that of WS$_2$, and no relative interlayer bandgap change would be allowed.

## Unconventional Stark effect

Interestingly, the average slope (indicated by black dashed line in Fig. 2b) of the redshift of $X_2$ is almost similar (about 5 meV/V) to that of the intra-layer WS$_2$ trion (X$^-$) or charged (XX$^-$) biexciton[31] (See Supplementary Fig. 6), but the average slope is higher for $X_1$ (-7 meV/V) and $X_0$ (-15 meV/V). The redshift of the intra-layer WS$_2$ trion emission peak with $V_g$ is directly related to the enhanced trion dissociation energy due to the extra energy required to place the remaining electron into the increasingly filled conduction band. Hence it can be correlated with the change in the Fermi energy due to doping[31–33]. This change is nearly equal to the shift in the WS$_2$ conduction band with respect to the WSe$_2$ valence band, making the average slopes of $X_2$ and WS$_2$ trion shift similar. This also is in agreement with the weak confinement of $X_2$.

However, the enhancement in the slope of the redshift for $X_1$ and $X_0$ cannot be explained from doping dependent interlayer bandgap reduction and suggests a strong additional effect of localization. To understand this further, we solve the Poisson equation to obtain the movement of bands with $V_g$ (see Supplementary Note 3 for the details of the calculation). The results are summarized in Fig. 2d. At small positive $V_g$, the bands shift downward in energy (middle panel, $V_g = 0.5$ V). However, at larger positive $V_g$, the central part of region I (right panel, $V_g > 0.5$ V) of the conduction band moiré well being

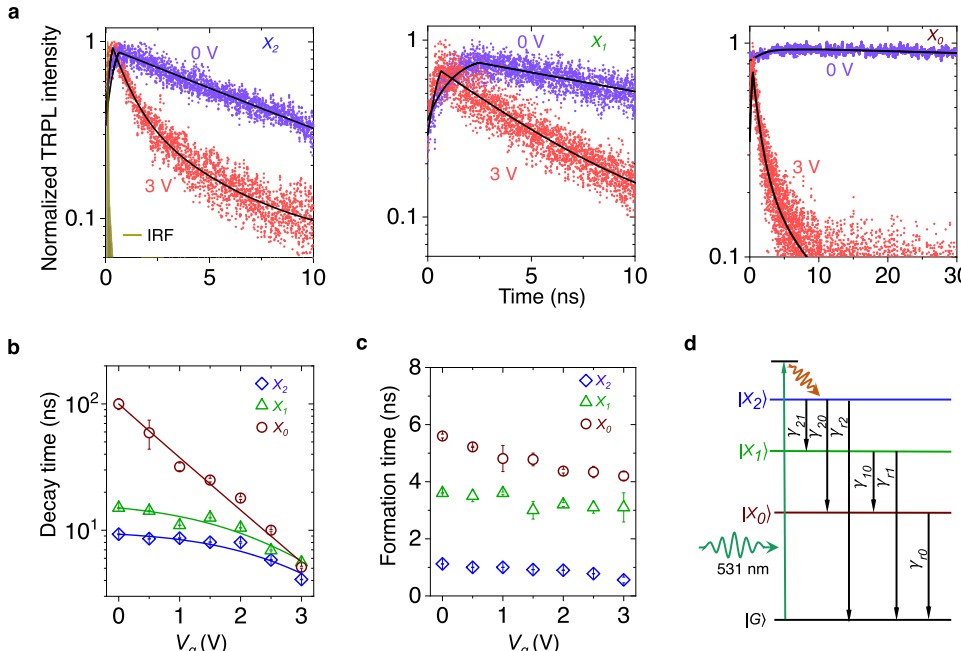

**Fig. 3 | Gate induced lifetime modulation of moiré exciton. a** Peak-resolved TRPL spectra (symbols) along with model (described in Methods) predicted fitting (black trace) at $V_g = 0$ and 3 V for $X_0$, $X_1$, and $X_2$. The IRF is shown in the left panel. **b** Extracted decay time (symbols) for different moiré ILEs as a function of $V_g$. Solid traces represent the model (Eq. (1)) prediction. **c** Extracted formation times plotted as a function of $V_g$. **d** Cascaded formation process for different ILEs, showing radiative ($\gamma_{r,i}$) channels for the exciton $X_i$ ($i = 0, 1, 2$), and inter-excitonic non-radiative paths ($\gamma_{ij}$) between excitons $X_i$ and $X_j$.

energetically closer to the Fermi energy supports more electron density than region II. Accordingly, due to the screening by the induced carrier density, region I starts moving down slower than region II. The net effect is a suppression in the local moiré fluctuation of the conduction band. Interestingly, the self-consistent electrostatics forces an amplification in the moiré potential fluctuation in the valence band of WSe₂: The suppressed movement of WS₂ bands in region I also reduces the movement of bands in WSe₂, while the stronger movement of WS₂ bands in region II (with relatively less carrier density) also pushes the WSe₂ bands more downward. The net result is a flattening of the electron moiré well in the WS₂ conduction band, causing a delocalization of the electron state, coupled with a deeper hole moiré well in the WSe₂ valence band, resulting in an enhanced localization of the hole state (zoomed in Fig. 2d, bottom panel). This modification of the moiré trapping potential, in turn, causes a reduction in the energy of the trapped electron state and an enhancement in the energy of the trapped hole state. The negative net change gives rise to an additional redshift in the localized exciton resonance ($X_0$ and $X_1$).

This results in an in-plane perturbation potential ($\Delta V$) with even parity about the high-symmetry points (Fig. 2e). $\Delta V$ is maximum at the center of the moiré well and reduces symmetrically away from the center. On the other hand, the wave function ($\psi$) has an even and odd parity for the ground ($X_0$) and first excited ($X_1$) states, respectively. This, in turn, results in a large (small) value of $|\psi_0|^2$ ($|\psi_1|^2$) around the center of the trap for $X_0$ ($X_1$), as shown in Fig. 2e. Due to such a strong overlap (non-overlap) of $\Delta V$ and $|\psi_0|^2$ ($|\psi_1|^2$), the first-order Stark effect ($\langle\psi|\Delta V|\psi\rangle$) is nonzero (negligible) for $X_0$ ($X_1$). Accordingly, we expect $X_0$ and $X_1$ to exhibit linear and parabolic Stark shift, respectively, with the in-plane local electric field ($\xi$), and hence with $V_g$, since our simulation suggests that $\xi$ is approximately linearly dependent on $V_g$ (see Supplementary Figure 7). Such local field effect will cancel out for the less-localized $X_2$ state. In Fig. 2f, the respective Stark shifts [$\delta_{X_{0,1}}(V_g) - \delta_{X_{0,1}}(V_g = 0)$ where $\delta_{X_0} = E_{X_2} - E_{X_0}$ and $\delta_{X_1} = E_{X_2} - E_{X_1}$] exhibit linear and parabolic variation with $V_g$ (reproduced in sample D4 as well, see Supplementary Fig. 8), in excellent agreement with the above analysis. We note that such Stark effect is unconventional since

the usual quantum-confined Stark effect (QCSE) in quantum wells, where the applied vertical electric field is uniform, results in a perturbing potential having odd parity. Thus the first-order QCSE (linear) is usually negligible, and we only observe a parabolic shift in the emission energy due to the second-order correction[34–38].

## Gate tunable exciton lifetime

Figure 3a shows the peak-resolved (spectral resolution of 0.8 meV) TRPL spectra (see "Methods" section) for $X_0$, $X_1$, and $X_2$, at $V_g = 0$ and 3 V, suggesting a faster decay at higher $V_g$ for all the ILE peaks. The transient response is captured well (solid black lines in Fig. 3a) by a set of rate equations and Gaussian formation model (see "Methods" section, Eqs. (3)–(5)). The extracted decay ($\tau_i$) and formation time ($\tau_{fi}$) are plotted for the exciton $X_i$, $i = 0, 1, 2$ in Fig. 3b, c. Around $V_g = 0$ V, the decay time varies over 10-fold from $X_0$ (∼100 ns) to $X_2$ (∼9 ns). However, at large $V_g$, all the three ILEs show similar decay time (4–6 ns). On the other hand, the formation times are relatively weaker function of $V_g$ and reduce slightly with increasing $V_g$.

The kinetics can be understood by the cascaded processes schematically depicted in Fig. 3d. At small $V_g$, the respective net lifetimes follow the trend $\tau_0 \gg \tau_1 > \tau_2$ (Fig. 3b), which is readily understood due to the additional non-radiative decay paths $\gamma_{20}$ and $\gamma_{21}$ for $X_2$, and $\gamma_{10}$ for $X_1$. The order of the respective formation times ($\tau_{f0} = 5.6$ ns, $\tau_{f1} = 3.6$ ns, and $\tau_{f2} = 1.1$ ns) in Fig. 3c, also supports the model of cascaded formation. In addition, a longer lifetime would mean the state is blocked for a longer duration, increasing the formation time.

The strong gate dependence of the ILE lifetime is captured through a simple model where the gate dependent non-radiative process is considered as proportional to induced carrier density (see Eqs. (6)–(7) in "Methods" section):

$$\tau_i(V_g) = \left[ \frac{1}{\tau_i(V_g = 0)} + C_i(e^{\alpha V_g} - 1) \right]^{-1} \quad (1)$$

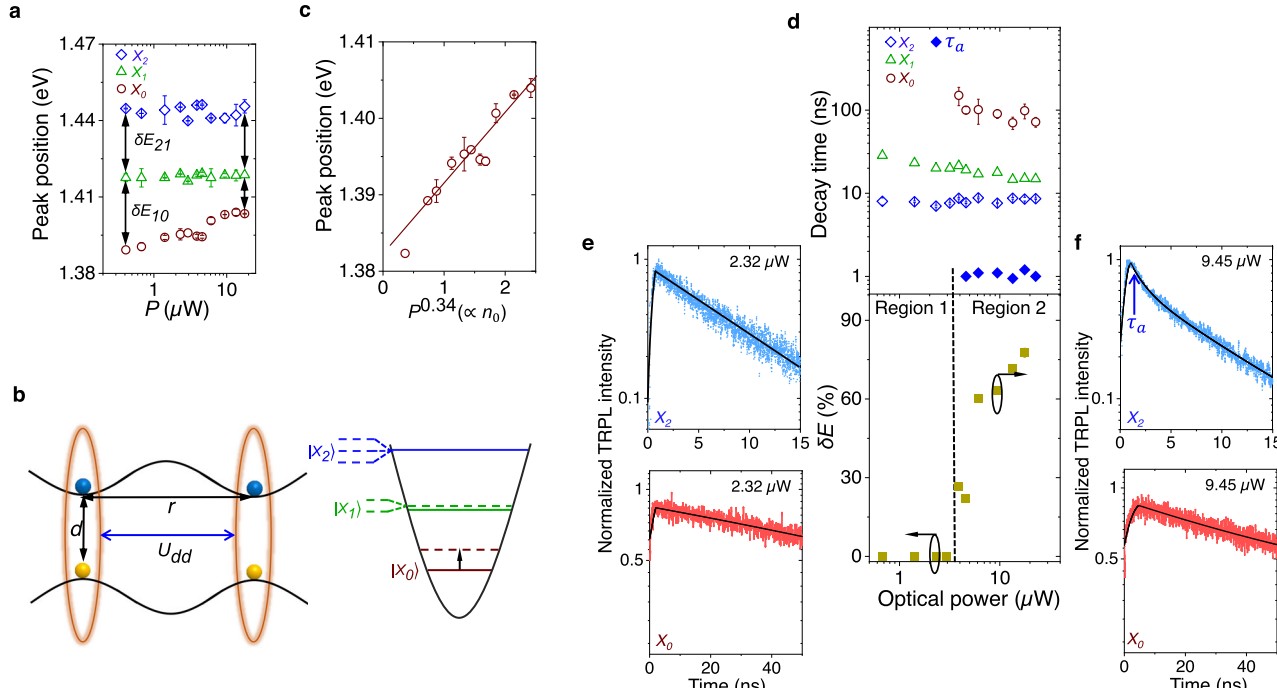

**Fig. 4 | Optical power dependent anharmonic tuning of moiré potential. a** PL peak position for $X_0$, $X_1$, and $X_2$, plotted against optical power ($P$). $X_0$ exhibits a strong blueshift (1 meV/μW) with $P$. The inter-excitonic peak separation is similar at low $P$, but becomes different at high $P$. **b** Left panel: Schematic representation of the interlayer excitonic dipole repulsion model. Right panel: Lifting of degeneracy for $X_2$ and $X_1$ in a two-dimensional harmonic oscillator shown schematically at higher $P$. Dipole repulsion results in blueshift of the states (dotted line), which is highest for $X_0$ (shown by a black arrow). **c** Peak position of $X_0$ (symbols) plotted against $P^{0.34}(\propto n_0)$, showing excellent linear fit. **d** Top panel: Extracted lifetime of $X_0$, $X_1$, and $X_2$ (in open symbols) plotted with optical power, showing a weak

dependence due to suppressed Auger process. The solid blue symbols ($\tau_a$) indicate additional decay path of $X_2$ due to anharmonicity induced degeneracy lifting at higher $P$. Bottom panel: Percentage change in the inter-exciton peak separation with $P$, indicating the degree of anharmonicity induced by $P$. The Regions 1 (harmonic) and 2 (anharmonic) are separated by a dashed black line, and correlates well with the appearance of $\tau_a$ in $X_2$. **e, f** The top and bottom panels show the TRPL spectra for $X_0$ and $X_2$, at (**e**) $P = 2.32$ and (**f**) 9.45 μW, respectively. $X_2$ decay becomes bi-exponential with a fast ($\approx 1$ ns) $\tau_a$ at higher $P$, while $X_0$ decay remains mono-exponential all through.

The model (solid traces in Fig. 3b) accurately reproduces the $V_g$ dependent lifetime values (symbols) by using $\alpha$ and $C_i$ as fitting parameters. We observe a $V_g$-modulation of $\tau_0$ by more than 20-fold from 100 to 5 ns (Fig. 3b), which correlates well with the PL intensity reduction of $X_0$ with $V_g$, in Fig. 2a. This is a direct evidence of the gate-induced non-radiative process due to the delocalization of the electron in the flattened conduction band (Fig. 2d). $X_0$ being the ground state of the well, the inter-excitonic transfer-related non-radiative decay channels (Fig. 3d) are suppressed. On the other hand, At low $V_g$, $\tau_1$, and $\tau_2$ are dominated by the (gate independent) non-radiative decay channels to other lower energy states (that is, $\gamma_{10}$, $\gamma_{20}$, and $\gamma_{21}$), hence remain nearly unchanged up to $V_g = 2$ V (Fig. 3b). The $V_g$-dependent non-radiative decay rate starts dominating only at large $V_g$ for $X_1$ and $X_2$, resulting in a reduction of $\tau_1$ and $\tau_2$.

**Optical power induced anharmonicity**
We now vary $P$ over nearly two decades using a pulsed laser (531 nm) at $V_g = 0$ V and plot the ILE peak positions in Fig. 4a. While $X_0$ exhibits a strong blueshift ($\approx 1$ meV/μW), the shift for $X_1$ and $X_2$ is negligible. Hence, the inter-excitonic separations ($\delta E_{21}$ and $\delta E_{10}$) do not remain equal at higher $P$, suggesting departure from harmonic behavior. Such anharmonicity and power-dependent blueshift can be understood by the perturbing potential ($U_{dd}$) arising from ILE dipolar repulsion[39,40]:

$$U_{dd} = \int n U(r) d^2 r = n \frac{q^2 d}{\epsilon_0 \epsilon_r} \quad (2)$$

where $U(r) = \frac{q^2}{2\pi\epsilon_0\epsilon_r}(\frac{1}{r} - \frac{1}{\sqrt{r^2+d^2}})$ is the repulsion between two ILE dipoles placed at a distance r (schematically shown in Fig. 4b, left panel). $\epsilon_0$ is the vacuum permittivity, $\epsilon_r$ is the effective relative permittivity of the heterojunction, $n$ is the effective concentration of exciton dipoles, and $d$ is the interlayer separation. Due to this induced anharmonicity, it is expected to observe a lifting of degeneracy for $X_1$ and $X_2$, as shown schematically in Fig. 4b (right panel). Since the lifetime of $X_0$ is significantly larger than that of $X_1$ and $X_2$, the steady-state density (generation rate × lifetime) of ILE dipoles is dominated by the population of $X_0$ ($n_0$). Since $I_{X0}(\propto n_0) \propto P^{0.34}$ (see Fig. 1h), Eq. (2) indicates that the blueshift ($E_{dd}$) of $X_0$ should follow $E_{dd} \propto P^{0.34}$, in good agreement with the linear fit in Fig. 4c. From Eq. (2), $n_0$ is estimated to be $\approx 9.5 \times 10^{11}$ cm$^{-2}$ (which is less than $N_M/2$) at the highest optical power used (17.7 μW).

To the best of our knowledge, the observed average rate of the blueshift with power for $X_0$ ($\approx 1$ meV/μW) is the highest reported value for ILE to date[39,41–43], indicating a strong inter-excitonic interaction. The strong confinement of $X_0$ does not allow it to drift out of the moiré trap in the presence of such dipole–dipole repulsion, resulting in a large blueshift. On the other hand, weaker confinement of $X_1$ and $X_2$ allows them to drift away under such dipolar repulsion, resulting in a suppressed blueshift in this small power regime.

Figure 4d, top panel (open symbols) shows the optical power dependent lifetime of $X_0$, $X_1$, and $X_2$. We notice that the lifetime for all the three species is a weak function of $P$. This is in stark contrast with intra-layer free exciton where Auger effect drastically reduces the lifetime at higher $P$[44,45]. Such a weak dependence of lifetime on $P$ is a result

of protection from Auger-induced exciton-exciton annihilation due to a combined effect of moiré trapping and strong dipolar repulsion.

For a perfect two-dimensional harmonic well, $X_0$, $X_1$, and $X_2$ are expected to exhibit a degeneracy of 1, 2, and 3, respectively. Through the optically induced anharmonicity, we expect the degeneracy of $X_1$ and $X_2$ to be lifted (Fig. 4b, right panel). However, our simulation suggests only < 2 meV fine-splitting, and the inhomogeneous broadening of the peaks does not allow us to observe such small splitting in the emission spectra.

Interestingly, while $X_2$ exhibits a mono-exponential decay at low power, its dynamics becomes bi-exponential at higher power ($P > 3.9\ \mu W$) with an additional lifetime of $\tau_a \sim 1\ ns$, as indicated by the blue solid symbols in Fig. 4d (top panel), and the TRPL spectra in the top panels of Fig. 4e, f. In the bottom panel of Fig. 4d, we quantify the degree of anharmonic perturbation by plotting, from Fig. 4a, the relative magnitude of the peak separation ($\delta E = \frac{\delta E_{21} - \delta E_{10}}{\delta E_{21}} \times 100\%$) with incident power (0% corresponding to the harmonic case). The strong correlation between the appearance of the faster additional decay (in region 2) and the strength of the anharmonic perturbation is evident. The faster additional decay likely arises from the fine-split higher energy state of $X_2$, which has reduced confinement into the moiré trap, thus having enhanced decay rate (schematically shown in Fig. 4b, right panel). Note that the decay of $X_0$ remains mono-exponential even at higher power since the ground state is non-degenerate (bottom panels of Fig. 4e, f).

In summary, we have shown that the exciton moiré potential in heterobilayer can be dynamically tuned through external stimuli, such as gate voltage and optical power. The usual harmonic approximation of moiré potential breaks down under such perturbation. The strength of such tunability is evidenced through moiré excitons exhibiting (a) confinement dependent tuning of features, (b) anomalous Stark shift where parity is reversed with respect to conventional quantum-confined Stark effect, (c) strong modulation of the lifetime and the inter-excitonic separation, and (d) a giant spectral blueshift through dipolar repulsion. The results will lead to intriguing experiments and applications exploiting dynamic tuning of moiré potential.

## Methods

### Device fabrication

We prepared the hBN-capped $WS_2/WSe_2$ heterojunctions using a sequential dry-transfer method (with micromanipulators) where the individual layers were exfoliated from bulk crystals (hq graphene) on polydimethylsiloxane (PDMS) using Scotch tape. For back-gated samples, the pre-patterned metal electrodes are prepared using photolithography followed by sputtering of Ni/Au (10/50 nm) and lift-off. The entire stack (for D1 and D4) is gated from the backside (from the $WS_2$ side) through hBN layer (dielectric) and the pre-patterned metal line. The $WS_2$ layer is contacted to a different electrode (Gr) for carrier injection. After completion of the transfer process, the devices are annealed inside a vacuum chamber ($10^{-6}$ mbar) at 250 °C for 5 h for better adhesion of the layers and removal of air bubbles. The angle and stacking between $WS_2/WSe_2$ layers are confirmed using SHG (see Supplementary Fig. 1).

### PL measurement

All the PL measurements on the samples are carried out in a closed-cycle cryostat at 4 K using a × 50 objective (0.5 numerical aperture) lense. The bottom gate voltages are applied using a Keithley 2636B source meter (for both PL and TRPL), and then the PL spectra are collected using a spectrometer with 1800 lines per mm grating and CCD (Renishaw spectrometer). We use the 532 nm CW and 531 nm pulsed lasers to excite the sample. The spot size for both pulsed and CW laser is ~ 1.5 μm. All the power values are measured using a silicon photodetector from Edmund Optics. All the error bars in different plots in the manuscript indicate mean ± standard deviation.

### TRPL measurement

Our custom-built TRPL setup comprises of a 531 nm pulsed laser head (LDH-D-TA-530B from PicoQuant) controlled by the PDL-800 D driver, a photon-counting detector (SPD-050-CTC from Micro Photon Devices), and a time-correlated single photon counting (TCSPC) system (PicoHarp 300 from PicoQuant). The pulse width of the laser is 40 ps. For the spectrally resolved TRPL from moiré ILEs, a combination of a long pass filter (cut in wavelength of 650 nm) and a wavelength-tunable monochromator (Edmund optics, 2 cm² Square holographic gratings) with 0.5 nm resolution (corresponding to about 0.8 meV resolution in the ILE spectral regime) are placed in front of the SPD. The peak position of the emission from ILEs are simultaneously measured along with TRPL measurement by performing in-situ PL (see Supplemental Material in ref. 31 for setup schematic). The instrument response function (IRF) has a full-width-at-half-maximum (fwhm) of 52 ps.

### Exciton formation and decay model

To fit the experimentally obtained TRPL data, we use three differential equations:

$$\frac{dn_0(t)}{dt} = f_0(t) - \frac{n_0(t)}{\tau_0} \qquad (3)$$

$$\frac{dn_1(t)}{dt} = f_1(t) - \frac{n_1(t)}{\tau_1} \qquad (4)$$

$$\frac{dn_2(t)}{dt} = f_2(t) - \frac{n_2(t)}{\tau_2} \qquad (5)$$

Here $n_i(t)$ is the time dependent population density, $\tau_i$ is the net decay time, and $f_i(t) = \frac{1}{\sigma_i \sqrt{2\pi}} e^{\frac{-(t-\tau_{fi})^2}{2\sigma_i^2}}$ is the Gaussian formation function, and $\tau_{fi}$ is the formation time measured from the laser excitation time for exciton $X_i$, $i = 0, 1, 2$. After solving these equations numerically, we fit the measured TRPL data from the three moiré exciton emissions using $\tau_{fi}$, $\sigma_i$, and $\tau_i$ as fitting parameters.

### Model for gate-voltage dependent lifetime

The net decay time ($\tau_i$) measured in TRPL (Fig. 3b), for exciton $X_i$ ($i = 0, 1, 2$) is given by:

$$\frac{1}{\tau_i(V_g)} = \frac{1}{\tau_{r,i}} + \frac{1}{\tau_{nr0,i}} + \frac{1}{\tau_{nrg,i}(V_g)} \qquad (6)$$

where $\tau_{r,i}$, $\tau_{nr0,i}$, and $\tau_{nrg,i}(V_g)$ represent the radiative lifetime, gate voltage independent non-radiative lifetime, and the gate voltage-dependent non-radiative lifetime, respectively. From Fig. 3d, $\frac{1}{\tau_{nr0,2}} = \gamma_{20} + \gamma_{21} + \gamma'_2$ for $X_2$, and $\frac{1}{\tau_{nr0,1}} = \gamma_{10} + \gamma'_1$ for $X_1$, and $\frac{1}{\tau_{nr0,0}} = \gamma'_0$, where $\gamma'_i$ is the rate of any other unaccounted non-radiative process for exciton $X_i$. Considering that the rate of the gate-dependent non-radiative process is proportional to induced carrier density, which in turn is an exponential function of $V_g$, we write $\frac{1}{\tau_{nrg,i}} = C_i e^{\alpha V_g}$, where $C_i$ and $\alpha$ are fitting parameters. By noting that $\frac{1}{\tau_{r,i}}$ is relative small (in Eq. (6)) and becomes smaller with an increase in $V_g$, we write

$$\frac{1}{\tau_i(V_g)} \approx \frac{1}{\tau_i(V_g = 0)} + C_i(e^{\alpha V_g} - 1) \qquad (7)$$

## Data availability
The data that support the findings of this study are available within the main text and Supplementary Information. Any other relevant data are available from the corresponding authors upon request.

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

## Acknowledgements
S.C. and K.M. acknowledge useful discussions with Garima Gupta, Nithin Abraham, Mayank Chhaperwal, and Manish Jain. K.W. and T.T. acknowledge support from the JSPS KAKENHI (Grant

Numbers 19H05790 and 20H00354). K.M. acknowledges the support from a grant from Science and Engineering Research Board (SERB) under Core Research Grant, a grant from the Indian Space Research Organization (ISRO), a grant from MHRD under STARS, and support from MHRD, MeitY, and DST Nano Mission through NNetRA.

## Author contributions

K.M. designed the experiment. M.D., S.C. and S.D. fabricated the devices and conducted the measurements. P.D. conducted the electrostatic simulation. R.B. and V.R. performed the SHG measurements for all samples. K.W. and T.T. grew the hBN crystals. S.C., M.D. and K.M. conducted the data analysis and wrote the manuscript with inputs from others.

## Competing interests

The authors declare no competing interests.
