## [Peer Review File · Nature Communications]

Reviewers' Comments:

Reviewer #1:

Remarks to the Author:

The manuscript titled "Harmonic to anharmonic tuning of moiré potential leading to unconventional Stark effect and giant dipolar repulsion in WS₂/WSe₂ heterobilayer" by Suman Chatterjee et al. reported the dynamic tuning of moiré potential in the WS₂/WSe₂ heterobilayer by gate voltage and optical power, which induces several interesting features of the moiré excitons such as the anomalous Stark shift, unequal inter-excitonic spectral separation, and giant lifetime tunability. The scientific problem and challenges in the related field were clarified clearly and the experimental data along with the analysis were well organized. Overall, this work provides a new perspective to study the moiré effect in 2D heterobilayers. Therefore, I recommend to publish it in this journal. The following issues should be addressed.

1. The authors should explain how to get the data of the calculated moiré potential shown in Figure 2c, either in the Method part or in the main text.
2. The authors mentioned that the 1D Poisson equation was used to obtain the movement of the bands with V_g . However, there were no details about this equation as well as no explanations of how to solve it to get the required information.
3. Meanwhile, it is difficult to understand the content related to Figure 2(d). The authors are recommended to provide a detailed illustration of this figure, as well as to clearly clarify the differences between the average slopes of X0, X1, and X2 under gate voltages.
4. The authors claimed that X0 features the linear Stark effect while X1 features the second-order Stark effect. However, it is really difficult to give this conclusion. Based on the data shown in Figure 2f, both of them are neither linear nor second-order.

Reviewer #2:

Remarks to the Author:

The paper "Harmonic to Anharmonic Tuning of Moiré Potential Leading to Unconventional Stark Effect and Giant Dipolar Repulsion in WS₂/WSe₂ Heterobilayer" by Suman Chatterjee et al. investigates the tuning of the moiré potential in a WS₂/WSe₂ moiré superlattice induced by a gate electric field and optical excitation power. The paper provides insights into the effects of the moiré potential on the optical properties of WS₂/WSe₂ and contributes to the field.

However, to further strengthen their results, the authors should consider reproducing their findings in an additional device. Relying on only one device is not convincing, especially since the results differ from those reported in the literature (see details below).

I would recommend the publication in Nature Communications if the reproducibility issue and the following questions can be resolved. This would increase the credibility of their results.

Below are some specific questions:

- 1) It has been reported that the moiré potential can be tuned by pressure and electric field. What is the advantage of these two anharmonic perturbations?
- 2) The authors observed three peaks in their study, whereas previous literature [such as Phys. Rev. Lett. 127, 037402 (2021), and many others] only reported one peak. What is the possible explanation for these differences, and has the sample quality improved? The authors may consider comparing the current results with the reported ones. There are several papers regarding the PL spectroscopy on the WS₂/WSe₂ moiré superlattices.
- 3) What method did the authors use to pick up the monolayer on the PDMS substrate, and is it true that exfoliating the flakes on the PDMS substrate produces better quality?

4) Why did the authors not observe the insulating states at one electron(hole) and two electrons(holes) per moiré superlattice site? This is not consistent with the reported literature. Reproducing one more device is helpful.

5) The authors have excluded possible alternate explanations such as phonon-sidebands, defect-bound exciton, and charged ILE based on their results of lifetime, power law, and tunability of the spectral separation of the peaks. Can they explain further why these effects have been excluded?

6) For the band alignment in Fig. 2b, why does the positive voltage have such a large effect? Can the red shift of the interlayer exciton be explained by the interaction between the interlayer exciton and the vertical electrical field? Also, is it possible that featureless on the hole doping side is due to the fact that the contact is only on WS₂ (Fig. 1c), and thus hole doping is hard due to the bad contact? It would be helpful to show the graphite contact in Fig. 1d.

7) The authors observed that the lifetime is tuned by the gate voltage. What is the relationship between the tuning of the moiré potential and the lifetime? Does this mean that at charge neutral, the moiré potential is deeper?

8) Why does the PL exciton intensity not scale linearly with the excitation power? What is the physical meaning of $\alpha_0 = 0.34$ and $\alpha_1 = 0.59$?

9) I suggest that the authors add the carrier density corresponding to the gate voltage in Fig. 2a. What is the electron doping at $V_g = 3$ V, by the way?

Reviewer #3:

Remarks to the Author:

This paper explores an important and timely topic in the field of moiré physics, namely the dynamic tuning of the moiré potential in a WS₂/WSe₂ heterobilayer through two anharmonic perturbations, induced by gate voltage and optical power. The authors demonstrate that a gate voltage can result in a local in-plane perturbing field with odd parity around the high-symmetry points, allowing for simultaneous observation of the first and second order Stark shift for the ground state and first excited state, respectively, of the moiré trapped exciton - an effect exactly opposite to conventional quantum-confined Stark shift. The second anharmonic tuning is demonstrated through exciton localization dependent dipolar repulsion, leading to an optical power-induced blueshift as high as 1 meV/microwatt, which is about 5 times higher than previous reports. The paper offers several intriguing features of the moiré excitons, including giant lifetime tunability, anomalous Stark shift, and dipolar repulsion induced large spectral blueshift.

The paper presents significant and novel results that advance the field of moiré physics. The experimental design and methods used are sound, and the data is presented clearly and comprehensively. However, there are a few key issues that should be addressed to further strengthen the paper.

Firstly, the lack of direct evidence for the existence of the Moiré potential is a major concern. While the author suggests that the near-equal inter-excitonic separation suggests the presence of harmonic Moiré potential wells, direct evidence in the form of TEM images of samples displaying Moiré patterns would be highly beneficial. Alternating circularly polarized photoluminescence is another method that could be employed to confirm the existence of Moiré excitons (Nature 2019, 567 (7746), 71-75).

Secondly, the author should provide several samples with different twist angles to explore the influence of different sizes of Moiré potential. This will help in better understanding the depth of the exciton Moiré potential and how it is affected by the twist angle and degree of lattice mismatch between the two heterobilayers.

Lastly, different stack layers of transition metal dichalcogenides (TMDCs) can significantly influence the Moiré potential energy, and the author did not specify the stack type in the article. To address

this issue, the author could prepare samples of two different types of stacks to explore the impact of interactions of interlayer excitons in a deeper level (Nature 610, 478–484 (2022)).

Overall, this paper is a valuable contribution to the field of Moiré physics, and addressing the aforementioned minor issues would further strengthen the paper.

We thank all the reviewers for their comments and suggestions. Below, we present a point-by-point response to all the comments. The changes in the manuscript are mentioned here, as well as highlighted in the text accordingly.

Reviewer #1:

The manuscript titled “Harmonic to anharmonic tuning of moiré potential leading to unconventional Stark effect and giant dipolar repulsion in WS₂/WSe₂ heterobilayer” by Suman Chatterjee et al. reported the dynamic tuning of moiré potential in the WS₂/WSe₂ heterobilayer by gate voltage and optical power, which induces several interesting features of the moiré excitons such as the anomalous Stark shift, unequal inter-excitonic spectral separation, and giant lifetime tunability. The scientific problem and challenges in the related field were clarified clearly and the experimental data along with the analysis were well organized. Overall, this work provides a new perspective to study the moiré effect in 2D heterobilayers. Therefore, I recommend to publish it in this journal.

Thank you!

The following issues should be addressed.

1. The authors should explain how to get the data of the calculated moiré potential shown in Figure 2c, either in the Method part or in the main text.

A description of the method to obtain Figure 2c is included in Supporting Note 2.

We formulated the depth of the moiré potential from the spectral separation of the interlayer excitonic (ILE) emission peaks (Figure 1e). The emission energies are at ≈ 25 meV separation from each other (1.392, 1.418 and 1.442 eV for X_0 , X_1 , and X_2 respectively). This near equal separation suggests a 2D harmonic oscillator well for moiré pockets, with energy separation of $\hbar\omega$ (25 meV) between each state. The ground state, X_0 , must be then at an energy $\hbar\omega$ above the bottom of the well, and this gives rise to a total well depth of $3\hbar\omega = 75$ meV (shown in the figure below), considering the upper most state is nearly delocalized, as suggested by our experimental results.

In order to plot this in real space (Figure 2c, main text), we use a periodic function such as $\Delta(r) = \sum_{\vec{n}=1}^6 V_n e^{i\vec{b}_n \cdot \vec{r}}$ (Ref. [6], Supporting note 2). The V_n and \vec{b}_n are the Fourier expansion coefficients of the moiré potential and 6 different reciprocal lattice vectors (in the moiré Brillouin zone), respectively. \vec{b}_n vectors are calculated from the monolayer lattice vectors, considering 4% lattice mismatch. A 75 meV well depth variation indicates a V_n of 21 meV. The 2D potential map is shown in Figure 2c (main text).

Changes in the text: We have added a paragraph in Supporting note 2 as: “The depth of the moiré potential is formulated from the spectral separation of the interlayer excitonic emission peaks (Figure 1e). The emission energies are at ≈ 25 meV separation from each other (1.392, 1.418 and 1.442 eV for X_0 , X_1 , and X_2 respectively). This near equal separation suggests an assumption of 2D harmonic oscillator well for moiré pockets is valid, with energy separation of $\hbar\omega$ (25 meV) between each state. The ground state, X_0 , must be then at an energy $\hbar\omega$ above the bottom of the well, and this gives rise to a total well depth of $3\hbar\omega = 75$ meV.”

2. The authors mentioned that the 1D Poisson equation was used to obtain the movement of the bands with V_g . However, there were no details about this equation as well as no explanations of how to solve it to get the required information.

To understand the electrostatics of the WS_2/WSe_2 moiré heterojunction, we numerically solve a 2D Poisson equation ($\nabla^2 \phi = \sigma \delta(z)$) where WS_2/WSe_2 heterobilayer is considered to be a 2D sheet of charge at $z = 0$ plane with a 2D charge density $\sigma = q(p - n + N)$. Here p , n are the 2D density

of electron and hole, respectively. N refers to the unintentional doping in the heterobilayer at zero gate voltage. Under positive gate voltage, $p \approx 0$, and

$$n(x) = N_{2D} \log \left[1 + \exp \left(\frac{E_F - E_C(x) + q\phi(x)}{k_B T} \right) \right]$$

where N_{2D} is the 2D density of states. E_F is the chemical potential and is referenced as zero. Both conduction (CB) and valence band (VB) profiles are spatially varying according to the spatial bandgap modulation in a moiré superlattice. Only CB (WS_2) and VB (WSe_2) profiles at K point [$E_C^K(x), E_V^K(x)$] corresponding to the direct bandgap are considered. The moiré potential for electrons and holes is assumed to be a simple cosine profile as: $E_C^K(x) = E_{C0}^K(x) + \Delta E_G^K \cos(k_x x)$ and $E_V^K(x) = E_{V0}^K(x) - \Delta E_G^K \cos(k_x x)$ with a spatially varying bandgap of $E_G^K = (E_{C0}^K - E_{V0}^K) + 2\Delta E_G^K \cos(k_x x)$. Here $k_x = 2\pi/a_M$, $a_M = 7.3$ nm is the moiré period, and $\Delta E_G^K = 75$ meV.

Changes in the text: The above discussion is added as Supporting Note 3, and referenced in main text as “To understand this further, we solve the 1D Poisson equation to obtain the movement of bands with V_g (see Supporting Note 3 for the details of the calculation).”

3. Meanwhile, it is difficult to understand the content related to Figure 2(d). The authors are recommended to provide a detailed illustration of this figure, as well as to clearly clarify the differences between the average slopes of X0, X1, and X2 under gate voltages.

Sorry for the confusion. For clarity, we now mention the slope values of the fitted redshift of each peak as an inset to the Figure 2b, as shown below:

All the three ILE states redshift due to interlayer band gap reduction depicted in Figure 2b, right panel. However, different slopes of red shift in the order $X_0 > X_1 > X_2$ can be explained by Figure 2d (right-bottom panel in the zoomed-in version).

Due to increasing V_g (n-doping), when the minimum of the conduction band (WS_2) moiré wells come close to the Fermi level, it supports higher electron density. This induces enhanced screening, which flattens the conduction band moiré well rapidly (shown in right Figure). On the other hand, to maintain self-consistent electrostatics forces (dictated by the Poisson equation), we find an amplification in the moiré potential fluctuation in the valence band of WSe_2 . These two effects reduce the net transition energy of the ILE.

The degree of confinement is also in the order $X_0 > X_1 > X_2$, which indicates X_0 will be the most affected state due to the flattening (deepening) of the conduction (valence) band moiré wells.

Changes in the text: The slope values are included in Figure 2b, as shown above. In addition, as suggested the Fig. 2d is illustrated in the caption with more details, as follows: “Top panel: Simulated conduction and valance band profile at three different V_g (0, 0.1, and 0.5 V) values obtained by solving the 1D Poisson equation with the moiré potential fluctuation (see Supporting Note 3 for details). For simulation, the thickness of the gate dielectric (hBN) is assumed to be 20 nm. Region I (II) in the top left panel denotes the minimum (maximum) energy of the WS_2

conduction band due to moiré potential induced spatial energy fluctuation. At lower V_g (top middle panel), the conduction band gradually comes down in energy towards the Fermi level (red dashed line) maintaining the same degree of fluctuation. At higher V_g (top right panel), when the conduction band is close to the Fermi level, it starts flattening due to screening. This also results in a deepening in the valence band fluctuation. Bottom panel: Zoomed-in Region I at $V_g = 0$ V (in left) and $V_g = 0.5$ V (in right). The transition energy for X_0 (E_{X_0} , shown by arrow) decreases at higher V_g .”

4. The authors claimed that X_0 features the linear Stark effect while X_1 features the second-order Stark effect. However, it is really difficult to give this conclusion. Based on the data shown in Figure 2f, both of them are neither linear nor second-order.

Figure 2f is again shown below with added error bars for clarity:

We agree with the reviewer that the fittings to the linear and parabolic curves are not perfect. This is primarily due to the experimental inaccuracy arising from the overall Stark shift in both cases. The Stark shift is obtained by subtracting the actual peak positions ($E_{X_2} - E_{X_0}$ and $E_{X_2} - E_{X_1}$). Nonetheless, we would like to note here that the second order (parabolic) Stark shift for the X_1 should be weaker compared with the first order one for X_0 . This is clearly evident from the figure above.

To verify our claim, we repeated the V_g dependent PL measurement in a different sample (D4), as shown below. This is in good agreement with the results obtained from the previous sample.

Changes in the text:

(a) We have added the error bars in Fig. 2f.

(b) The new Stark shift plot obtained from sample D4 is added in Supporting Figure 8, and referenced in main text as “In Figure 2f, the respective Stark shifts $[\delta_{X_{0,1}}(V_g) - \delta_{X_{0,1}}(V_g = 0)]$ where $\delta_{X_0} = E_{X_2} - E_{X_0}$ and $\delta_{X_1} = E_{X_2} - E_{X_1}$] exhibit linear and parabolic variation with V_g (reproduced in sample D4 as well, see Supporting Figure 8), in excellent agreement with the above analysis.”

Reviewer #2:

The paper “Harmonic to Anharmonic Tuning of Moiré Potential Leading to Unconventional Stark Effect and Giant Dipolar Repulsion in WS₂/WSe₂ Heterobilayer” by Suman Chatterjee et al. investigates the tuning of the moiré potential in a WS₂/WSe₂ moiré superlattice induced by a gate electric field and optical excitation power. The paper provides insights into the effects of the moiré potential on the optical properties of WS₂/WSe₂ and contributes to the field.

Thank you!

However, to further strengthen their results, the authors should consider reproducing their findings in an additional device. Relying on only one device is not convincing, especially since the results differ from those reported in the literature (see details below).

I would recommend the publication in Nature Communications if the reproducibility issue and the following questions can be resolved. This would increase the credibility of their results.

Thanks for your comment. We have reproduced the multiple- peak features of the ILE emission in another sample (D2) having twist angle (θ) of $54 \pm 1^\circ$ (confirmed from SHG data, Supporting Figure 1). Also, to elucidate the systematic twist angle dependence, we fabricated a third sample (D3) with large misalignment. The PL spectra from D1 (original sample presented in the main manuscript), D2 and D3 are shown below:

Clearly, like D1, D2 features equally separated (~ 9 meV) ILE peaks, however, with smaller separation due to larger misalignment (and hence shallower moiré well). On the other hand, D3, with its large misalignment, features a single broad emission peak.

In addition, we have repeated the Stark effect measurement in another sample (D4) [which also exhibits 3-peak ILE emission feature] as shown below:

The Stark effect data above is in good agreement with the results obtained from sample D1 [Figure 2f of main text], that is, X_0 and X_1 exhibiting linear and parabolic Stark shift, respectively.

Changes in the text:

(a) The above comparison of stacking angle dependent PL emission from different samples is included in Supporting Figure 3 along with the SHG data in Supporting Figure 1.

(b) The new Stark shift plot obtained from sample D4 is included in Supporting Figure 8, and referenced in main text as “In Figure 2f, the respective Stark shifts $[\delta_{X_{0,1}}(V_g) - \delta_{X_{0,1}}(V_g = 0)]$ where $\delta_{X_0} = E_{X_2} - E_{X_0}$ and $\delta_{X_1} = E_{X_2} - E_{X_1}$] exhibit linear and parabolic variation with V_g (reproduced in sample D4 as well, see Supporting Figure 8), in excellent agreement with the above analysis.”

Below are some specific questions:

1. It has been reported that the moiré potential can be tuned by pressure and electric field. What is the advantage of these two anharmonic perturbations?

All these different stimulations perturb the system in different ways. For example, applying pressure would change the lattice constant and hence the band structure, while an electric field [we mean, a vertical electric field with zero common mode voltage (i.e., zero doping)] modifies the bands and creates a Stark shift.

On the other hand, both the stimulations explored in this work [that is, electron density (doping) and exciton density (optical power)] perturb the system through many-body effects.

From specific advantage point of view, we believe that compared with pressure induced modulation, techniques such as electric field, gate voltage induced doping, and optical excitation have the advantage of being dynamic (and fast) in nature and could be more suited for practical device applications.

2. The authors observed three peaks in their study, whereas previous literature [such as Phys. Rev. Lett. 127, 037402 (2021), and many others] only reported one peak. What is the possible explanation for these differences, and has the sample quality improved? The authors may consider comparing the current results with the reported ones. There are several papers regarding the PL spectroscopy on the WS₂/WSe₂ moiré superlattices.

Thank you for providing the reference, which we have included as Ref. [23] in the main text.

We would like to note that both single peak and multiple peaks have been reported in WS₂/WSe₂ moire samples. Please see, for example, a paper by Xueqian Sun *et al.* (Ref. [28], Nature, 610(7932):478-484, 2022). This paper observes multiple peaks from a suspended WS₂/WSe₂ stack.

The observation of single versus multiple peaks depends upon many parameters, such as: sample quality (such as sample inhomogeneity and adhesion between two layers), type of stacking (AA or AB), stacking angle (or, moiré well depth), residual doping, and measurement conditions (such as, gate voltage, optical power).

As already stated in response to the reproducibility comment earlier, in the revised manuscript, we now show a systematic dependence of varying stacking angle on the ILE emission peak feature (see below as well).

In any case, as mentioned earlier, twist angle is not the only parameter that can provide multiple versus single peaks. For example, as mentioned in the main manuscript, the different peaks obey different power law, and the broadening also increases with power, hence the number of peaks in the PL spectra can vary with the incident optical power.

Changes in the text:

(a) The reference provided by the reviewer and the paper by Xueqian Sun *et al* is cited as Refs. [23] and [28] in main text, respectively.

(b) We also included the above twist angle dependent spectra in the Supporting Figure 3. This is referenced in main text as “This inter-excitonic separation can be tuned by varying the twist angle, which regulates the depth of the moiré potential well [4,30]. We verified this by measuring twist angle dependent PL spectra from three samples [D1 (59°), D2 (54°) and D3 (large angle misalignment)] in Supporting Figure 3.”

3. What method did the authors use to pick up the monolayer on the PDMS substrate, and is it true that exfoliating the flakes on the PDMS substrate produces better quality?

We use a dry-transfer technique to prepare the hBN-capped WS₂/WSe₂ heterojunction. First, we prepare material tapes (using scotch tape 3M™) from the crystal, then we directly exfoliate from the tape to the PDMS sheet (Gel-Pak). The monolayers are identified on the PDMS sheet by optical contrast using a Leica microscope and transferred layer by layer (under the microscope) on the Au back gate (see Methods sample preparation section). The few layers of hBN and graphene (electrode) are also transferred following the same process.

In a separate work (unpublished), we prepared a hBN capped WS₂/WSe₂ heterojunction, where we picked up individual mono and hBN layers from a blank Si/SiO₂ substrate using spin-coated PPC (dissolved in Anisole)-PDMS stamp. After preparation, the full stack is dropped (by regulating the substrate temperature) onto an Au back gate. However, we do not observe any significant quality difference between the samples prepared by these two techniques in terms of optical quality.

4. Why did the authors not observe the insulating states at one electron (hole) and two electrons(holes) per moiré superlattice site? This is not consistent with the reported literature. Reproducing one more device is helpful.

Our calculated moiré trap density is $\sim 2 \times 10^{12} \text{ cm}^{-2}$ (for moiré period of 7.3 nm), and the estimated n-doping density at the highest $V_g (=5 \text{ V})$ is $\sim 1.5 \times 10^{12} \text{ cm}^{-2}$ [Note that, this is likely overestimated as we are neglecting the density-of-states capacitance of the heterojunction, explained in more details in response to comment #9]. Accordingly, we did not achieve one electron per moiré site in our experiment even with highest gate voltage applied. Since such insulating states are beyond the scope of the work, we did not increase the gate voltage further.

Changes in the text: The maximum filling is mentioned in Gate tunability section as “The estimated n-doping density at the highest applied $V_g (= 5 \text{ V})$ is $< 1.5 \times 10^{12} \text{ cm}^{-2}$ (see Supporting Figure 4). This is well below the moiré trap density of $n_0 \approx 2 \times 10^{12} \text{ cm}^{-2}$ for $a_M \sim 7.3 \text{ nm}$.”

5. The authors have excluded possible alternate explanations such as phonon-sidebands, defect-bound exciton, and charged ILE based on their results of lifetime, power law, and tunability of the spectral separation of the peaks. Can they explain further why these effects have been excluded?

Phonon sidebands: We observe nearly equal spacing between our peaks in the experiment. Interestingly, both harmonic quantum well and phonon sidebands would provide equally spaced peaks. However, had they been arising from phonon side-bands, the spacing would remain the same even at higher optical power or at higher doping, which is not the case in our experiments. Accordingly, we can rule out the possibility of their origin through phonon side-bands.

Defect bound excitons: Defect bound excitons are usually of broad linewidth, and the line shape does not get significantly modulated with power/electrical bias. In addition, defect luminescence usually saturates quickly with an increment in the incident power. In our experiment, we observe high quality features in the luminescence spectra, with the highest energy peak showing a power law of 1. In addition, the high tunability of the features of the peaks by gate voltage and optical power, along with their relative separation help us to exclude the possibility of the defect bound excitons.

Charged ILE: As shown in Figure 1a, the intensity of all the ILE peaks reduces with an increase in the n-doping, which agrees well with charge neutral exciton. On the other hand, according to reports of charged ILE, their luminescence intensity increases with higher doping, and hence their presence in our observation can be ruled out.

In addition, the alternating signs of the degree of circular polarization (added as Supporting Figure 2) also help us to rule out these alternate possibilities.

Changes in the text: We change the relevant text as:

“...Possible alternative explanations, such as phonon-sidebands and defect-bound excitons, are unlikely in our samples based on the observations including alternating signs of the DOCP and systematic tuning of the ILE peak separation with twist angle, doping, and optical power (discussed later).”

“...The reduction in emission intensity with an increase in V_g rules out the charged excitonic (trion) nature of any of the three peaks.”

6. For the band alignment in Fig. 2b, why does the positive voltage have such a large effect? Can the red shift of the interlayer exciton be explained by the interaction between the interlayer exciton

and the vertical electrical field? Also, is it possible that featureless on the hole doping side is due to the fact that the contact is only on WS₂ (Fig. 1c), and thus hole doping is hard due to the bad contact? It would be helpful to show the graphite contact in Fig. 1d.

Yes, the reviewer rightly pointed out that a large part of the redshift of interlayer emission peaks is due to the bottom gate vertical field (the other part arising from the Stark effect as explained in the paper). Please refer to the right panel of Fig. 2b in the main manuscript.

As suggested, we show below the graphite contact on WS₂. The reviewer is also right about the featureless negative V_g side, since it is difficult to dope WS₂ p-type through electrostatics (see, for example, our previous work: Murali *et al.*, *Advanced Functional Materials*, 31, 2010513, 2021).

Changes in the text: Figure 1d is changed accordingly showing the graphite contact.

7. The authors observed that the lifetime is tuned by the gate voltage. What is the relationship between the tuning of the moiré potential and the lifetime? Does this mean that at charge neutral, the moiré potential is deeper?

We have explained this through Fig. 2d. The moiré potential for electron is indeed deeper at the charge neutral point (Figure 2d, left-bottom panel). At higher n-doping, the conduction (valance) band moiré well flattens (deepens), as shown in Figure 2d (main text). This conduction band flattening causes a delocalization of the electron. This has two effects: (i) This decreases the overlap between electron and hole wavefunction, causing a reduction in the radiative decay rate. (ii) The delocalization of the electron causes a stronger non-radiative decay rate.

We modeled the V_g dependent lifetime governed by the above mechanisms as follows (equation 1 in main text): $\tau_i(V_g) = \left[\frac{1}{\tau_i(V_g=0)} + C_i(e^{\alpha V_g} - 1) \right]^{-1}$ which nicely captures our experimental data (shown below, Fig. 3b in main text).

8. Why does the PL exciton intensity not scale linearly with the excitation power? What is the physical meaning of $\alpha_0 = 0.34$ and $\alpha_1 = 0.59$?

The smaller-than-one exponent in the power law indicates a blockade in the generation of the exciton with an increase in the power law. Such blockade, in the context of current work, arises from two reasons: (a) moiré-trapping induced reduced number of states, and (b) long lifetime induced blockade. Moiré-trapped excitons would block creation of more inter-layer excitons in these states during its lifetime. Hence, this causes an effective blocking, and the generation rate of moiré-trapped exciton cannot catch up with the rate at which the optical power increases. Accordingly, the power law exponent reduces with an increase in the lifetime of the moiré exciton, and we observe an exponent of 0.34, 0.59, and ~ 1 for X_0 , X_1 , and X_2 , respectively.

9. I suggest that the authors add the carrier density corresponding to the gate voltage in Fig. 2a. What is the electron doping at $V_g = 3$ V, by the way?

The charge carrier density can only be accurately estimated once we know the threshold voltage (and hence the overdrive amount: $V_g - V_t$). Since the device did not have two contacts (for source and drain) to measure the drain current, we cannot comment on V_t , and hence we cannot provide an accurate estimate of the carrier density. However, assuming $V_t = 0$ V, hBN dielectric constant as 3.03, based on the hBN thickness used, we can roughly estimate the carrier density as plotted in the Figure below.

We must also keep in mind that such an estimate is based on $C_{hBN} * V_g$, which completely ignores the semiconductor (WS_2/WSe_2) channel capacitance (C_s) [also known as quantum capacitance or density of states capacitance] that comes in series with the C_{hBN} , and thus only close to accurate when C_s is much higher than C_{hBN} (which happens at large V_g). The system is not efficiently doped in the $V_g < 0$ V (p-doping) side, so this doping-density calculation is not valid in that regime (hence not shown here).

Based on this estimate the doping density at 3 V is $9 \times 10^{11} \text{ cm}^{-2}$.

Changes in the text: The above figure along with doping density is included in Supporting Figure 4 and referenced in the main text as “The estimated n-doping at the highest applied $V_g (= 5 \text{ V})$ is

$< 1.5 \times 10^{12} \text{ cm}^{-2}$. This is well below the moiré trap density of $n_0 \approx 2 \times 10^{12} \text{ cm}^{-2}$ for $a_M \sim 7.3 \text{ nm}$ (see Supporting Figure 4).”

Reviewer #3:

This paper explores an important and timely topic in the field of moiré physics, namely the dynamic tuning of the moiré potential in a WS₂/WSe₂ heterobilayer through two anharmonic perturbations, induced by gate voltage and optical power. The authors demonstrate that a gate voltage can result in a local in-plane perturbing field with odd parity around the high-symmetry points, allowing for simultaneous observation of the first and second order Stark shift for the ground state and first excited state, respectively, of the moiré trapped exciton - an effect exactly opposite to conventional quantum-confined Stark shift. The second anharmonic tuning is demonstrated through exciton localization dependent dipolar repulsion, leading to an optical power-induced blueshift as high as 1 meV/microwatt, which is about 5 times higher than previous reports. The paper offers several intriguing features of the moiré excitons, including giant lifetime tunability, anomalous Stark shift, and dipolar repulsion induced large spectral blueshift.

The paper presents significant and novel results that advance the field of moiré physics. The experimental design and methods used are sound, and the data is presented clearly and comprehensively.

Thank you!

However, there are a few key issues that should be addressed to further strengthen the paper.

1. Firstly, the lack of direct evidence for the existence of the Moiré potential is a major concern. While the author suggests that the near-equal inter-excitonic separation suggests the presence of harmonic Moiré potential wells, direct evidence in the form of TEM images of samples displaying Moiré patterns would be highly beneficial. Alternating circularly polarized photoluminescence is another method that could be employed to confirm the existence of Moiré excitons (Nature 2019, 567 (7746), 71-75).

Thanks for this comment. We have measured the degree of circular polarization (shown below, spectra in top panel and degree of polarization in bottom panel) for the ILE emission with a 705 nm laser excitation (spectrally close to WSe₂ intralayer excitonic emission) at 4K. The alternate sign of circular polarization of the peaks is clear, albeit the degree of polarization is small. This indicates the moiré-excitonic nature of the peaks.

Changes in the text: The above Figure indicating existence of moiré pattern is included as Supporting Figure 2. We change the relevant text in the main manuscript as: “The peaks exhibit alternating sign of the degree of circular polarization (Supporting Figure 2), indicating the existence of moiré superlattice [4,6,29].”

2. Secondly, the author should provide several samples with different twist angles to explore the influence of different sizes of Moiré potential. This will help in better understanding the depth of the exciton Moiré potential and how it is affected by the twist angle and degree of lattice mismatch between the two heterobilayers.

Our original sample had a twist angle of $\sim 59^\circ$. We have prepared two new samples (D2 and D3). D2 has a twist angle of $54 \pm 1^\circ$, whereas D3 is significantly misaligned. Below we present twist angle dependent PL emission from all these three samples with 532 nm excitation at 4K.

Clearly, D1 and D2 exhibit equally spaced ILE emission peaks, with the separation being higher in D1 (~ 25 meV) than D2 (~ 9 meV). This can be attributed to deeper moiré potential in D1.

On the other hand, in D3 (with large twist angle), we observe a single emission peak, likely due to shallow moiré wells, and inhomogeneity induced by large moiré-pocket density.

Changes in the text: We have included the above twist angle dependent spectra in the Supporting Figure 3. This is referenced in main text as “This inter-excitonic separation can be tuned by varying the twist angle, which regulates the depth of the moiré potential well [4,30]. We verified this by measuring twist angle dependent PL spectra from three samples [D1 (59°), D2 (54°) and D3 (large angle misalignment)] in Supporting Figure 3.”

3. Lastly, different stack layers of transition metal dichalcogenides (TMDCs) can significantly influence the Moiré potential energy, and the author did not specify the stack type in the article. To address this issue, the author could prepare samples of two different types of stacks to explore the impact of interactions of interlayer excitons in a deeper level (Nature 610, 478–484 (2022)).

We would like to note that we already mentioned in the original manuscript that our twist angle is about 59° , which indicates that our stacking type to be hexagonal (AB type). This is confirmed from the SHG color plot (Figure 1a, Supporting Information), as shown below. The measured

junction area (WS_2/WSe_2) on back-gate shows a sharp drop in SHG intensity with respect to the surrounding regions, which is a signature of AB stacking (Ref. [4,5]).

We made all the samples with a twist angle around 60° (AB stacking) as this stacking type is known to have stronger interlayer coupling (Ref. [8,10,11]).

Overall, this paper is a valuable contribution to the field of Moiré physics, and addressing the aforementioned minor issues would further strengthen the paper.

Reviewers' Comments:

Reviewer #1:

Remarks to the Author:

The authors have addressed all the questions properly, and made corresponding corrections in the revised version.

Reviewer #2:

Remarks to the Author:

Thanks to additional experimental works and the completeness of the answers to my comments.

I recommend it for publication in Nature Communications.

Reviewer #3:

Remarks to the Author:

The authors have thoroughly addressed my inquiries and supplemented the research with additional experimental data. The paper showcases remarkable and groundbreaking findings that significantly contribute to the field of moiré physics. The experimental design and methodologies employed are robust, while the data is presented in a lucid and comprehensive manner. I am delighted to endorse this manuscript for publication in Nature Communications.